# A PRPD-Based UHF Filtering and Noise Reduction Algorithm for GIS Partial Discharge

**DOI:** 10.3390/s23156763

**Published:** 2023-07-28

**Authors:** Weixing Fang, Guojin Chen, Wenxin Li, Ming Xu, Wei Xie, Chang Chen, Wanqiang Wang, Yucheng Zhu

**Affiliations:** 1College of Mechanical Engineering, Hangzhou Dianzi University, Hangzhou 310018, China; farrfang@163.com (W.F.); chenguojin@163.com (G.C.); 41096@hdu.edu.cn (M.X.); 41439@hdu.edu.cn (C.C.); 40273@hdu.edu.cn (W.W.); zhuyucheng202205@163.com (Y.Z.); 2Hangzhou Kelin Electric Co., Ltd., Hangzhou 310011, China; xiewei@klec.com.cn; 3Anji Intelligent Manufacturing Technology Research Institute Co., Ltd., Hangzhou Dianzi University, Huzhou 313000, China

**Keywords:** partial discharge, PRPD, secondary noise reduction, sensitivity

## Abstract

The online detection of partial discharge (PD) in gas-insulated switchgear (GIS) is a crucial and powerful tool for maintaining their reliability. However, extracting weak discharge signals from strong disturbances is a significant challenge. The presence of noise can hamper the identification and localization of PD types, making the extraction of pure PD signals the focus of current research. This paper proposes a PRPD-based PD filtering algorithm that analyzes interference using the output information from PRPD and sets threshold parameters for noise reduction processing. This method is mainly used for secondary noise reduction at a later stage, without analyzing the noise source, to achieve effective signal acquisition while retaining the characteristics of the PD signals, thereby improving the system’s sensitivity and the signal’s purity.

## 1. Introduction

Partial discharge is an early manifestation of insulation failure in gas-insulated switchgear (GIS) and can cause damage and aging of the insulation [1]. The emergence of partial discharge is due to various factors such as insulation material, product design, production processing and manufacturing, sharp corners and burrs in parts, surface moisture, and poor metal structural part contact [2,3,4,5]. This paper focuses on five typical types of PD observed in GIS: tip discharge, particle discharge, air gap discharge, surface discharge, and suspension discharge. A comprehensive understanding of these types of PD is essential for the proper maintenance of GIS and to prevent insulation failures.

According to the principles related to the distribution of electromagnetic fields around the power equipment’s partial discharges (PDs), each PD is accompanied by a steep current pulse with a frequency of 300 MHz to 3 GHz, which radiates electromagnetic waves to the surrounding area at the nanometer level [6]. The testing site contains numerous pieces of power equipment, resulting in a complex actual situation with multiple sources of interference. Common sources of interference include power supply interference, various forms of electromagnetic interference, poor contact of the test circuit or the test equipment’s discharge, interference of the grounding system, and potential discharge of suspended metal objects. Furthermore, interference can enter the monitoring system through the sensor along with the PD signals [7]. Removal or reduction of PD signal interference is crucial to improving the accuracy and sensitivity of PD detection. The principle of noise reduction can be divided into three parts: the source of interference, the interference path, and late signal noise reduction. Among them, late signal noise reduction is the mainstream research direction, which does not require a detailed analysis of different equipment and environments based on the causes of interference.

Extensive research has been conducted to address the elimination and suppression of interference. In the literature [8,9], various digital filtering techniques such as high-pass and low-pass filters, adaptive filters, etc., have been explored. Filters are simple and widely used. High-pass filters effectively remove low-pass signals, and low-pass filters remove high-pass signals. The disadvantage, however, is that they are prone to unclean noise processing. The literature [10] investigated the use of the short-time Fourier transform (STFT), which involves applying a window to the signal to enable segmented Fourier transforms and facilitate the processing of non-smooth signals. However, the method has certain prerequisites, which are demanding and require certain stability and repeatability of the signal. In [11], the application of fast Fourier transforms (FFTs) and artificial neural networks was examined to reduce PD noise, with the artificial neural network demonstrating a remarkable signal-to-noise ratio. However, it produces peaks, thus affecting the signal-to-noise ratio, and does not perform well with low-frequency noise. Additionally, the literature [12] focused on estimating the time delay of analog signals under interference and proposed a time delay estimation algorithm based on fourth-order cumulants and the bispectrum. This has the significant advantage of being insensitive to Gaussian noise with unknown correlation properties. The disadvantages are the need for precise time synchronization and high sampling rate signal acquisition, as well as the need to solve a system of nonlinear equations, which is computationally complex.

Wavelet transforms, as a mainstream denoising method developed from Fourier transforms, have been widely studied [13,14,15]. Wavelet noise reduction typically employs hard or soft thresholding methods, where the signal after hard thresholding becomes coarse and more prone to oscillation. Conversely, the signal after soft thresholding is relatively smooth with good continuity but may have a constant deviation [16]. Furthermore, in PD noise reduction, wavelet coefficients only consider signal strength without considering time-domain characteristics. This can result in the loss of PD signal details and severe distortion. Additionally, complex signals may introduce errors in noise reduction processing. These issues can impact the reliability of the signal. Considering various environmental, equipment, and material limitations, complete noise elimination based on retaining signal characteristics is not achievable. Therefore, secondary noise reduction processing of the signal is necessary.

This paper proposes a late secondary noise reduction filtering method for gas-insulated switchgear (GIS) based on conventional noise reduction techniques (filters and wavelets), along with a PD denoising algorithm using PRPD (phase-resolved partial discharge). Previous studies on PRPD have mainly focused on identifying discharge patterns [17,18], analyzing PRPD to discriminate discharge causes [19] and achieving crosstalk suppression through PRPD image noise reduction [20] by extracting dominant ANGPD (adjacent to noise gap partial discharges) clouds via two independent filtering paths. But the method requires preprocessing of the image, and the effectiveness of the preprocessing is affected by the quality of the image. At the same time, the histogram features can be disturbed by random variables. In this paper, we primarily utilize the information such as the rising edge, pulse width, amplitude, synchronous count, and effective signal of the pulse signal obtainable from PRPD to perform the noise reduction process. By inputting the PRPD valid signal and setting two parameters, the interference is analyzed to reduce noise without the need for complex calculations. This can be compared with the frequency and amplitude of periodic narrowband noise computed using the sub-matrix maximum modulus method and suppressed by an inverse phase cancellation technique, as presented in the literature [21]. Also, UHF PD signals in Gaussian white noise are suppressed by using the singular value decomposition denoising method. The method is simpler and does not require complex calculations. 

## 2. Phase-Resolved Partial Discharge (PRPD)

Feature extraction for PD signals mainly includes phase-resolved partial discharge (PRPD), phase-resolved pulse sequence (PRPS), time-resolved partial discharge (TRPD), and voltage difference (Δu) [22]. PRPD mapping is a two-dimensional scatter plot consisting of three representations: apparent discharge (q), discharge phase (φ), and discharge frequency (n). In general, the *x*-axis represents the phase, the *y*-axis represents the signal intensity, and more points indicate higher discharge pulse density; PRPS mapping is a real-time three-dimensional bar graph, the most critical UHF method for type identification analysis mapping. Typically, the *x*-axis indicates the UHF phase, the *y*-axis indicates the UHF period, and the *z*-axis indicates the amplitude (or signal strength), which can quickly diagnose the insulation status of the equipment; TRPD is a time distribution analysis of the PD pulse signal to analyze the signal characteristics and parameters for pattern recognition; the $\Delta u_{n}$ distribution is a Δu pattern, obtained from the PD pulse sequence, indicating the reference voltage difference between two adjacent discharge pulses Δu’s distribution characteristics. PRPD is currently the most commonly used with stable performance among them.

The traditional approach for PRPD-based feature volume extraction is to introduce five normal distribution statistical operators [23,24]: skewness $S_{k}$, kurtosis $k_{u}$, phase asymmetry ψ, discharge asymmetry Q, and cross-correlation factor $C_{C}$. The formula is as follows:

Skewness $S_{k}$: Reflects the left-right skew of the graph shape concerning the normal distribution ($S_{k}$ > 0 left skew, $S_{k}$ = 0 symmetric, $S_{k}$ < 0 right skew):(1)$$S_{k}=\frac{\sum {\left(x_{i}-\mu \right)}^{3}}{\sigma^{3}},$$
where $x_{i}$ is the phase of the ith phase window; $P_{i}$, μ, and σ denote the probability, mean, and standard deviation of the occurrence of events in phase window i, respectively.

Kurtosis $k_{u}$: The degree of protrusion of the data relative to the shape of the ortho distribution (the steepness of the ortho distribution is 0, $k_{u}$ > 0 plot profile is sharper and steeper than the ortho distribution, and $k_{u}$ < 0 plot profile is flatter than the ortho distribution profile):(2)$$k_{u}=\frac{\sum {\left(x_{i}-\mu \right)}^{4}\cdot P_{i}}{\sigma^{4}}-3,$$
where the meaning of the parameter is the same as Equation (1).

Phase asymmetry ψ: The difference in the onset phase of the discharge in the positive and negative half of the φ−q spectrum:(3)$$\psi =\frac{\varphi_{in}^{-}}{\varphi_{in}^{+}},$$
where $\varphi_{in}^{-}$, $\varphi_{in}^{+}$ are the starting phase angles of the discharge in the positive and negative half circumference of the φ−q spectrum, respectively.

Discharge asymmetry Q: The difference in the average discharge in the positive and negative half-weeks of φ−q spectra:(4)$$Q=\frac{Q_{s}^{-}/N^{-}}{Q_{s}^{+}/N+},,$$
where $Q_{s}^{-}$, $Q_{s}^{+}$ are the sums of positive and negative half-cycle discharges of φ−q spectra, and $N^{-}$, $N^{+}$ are the sums of positive and negative half-cycle discharges of φ−q ranges, respectively.

The cross-correlation factor $C_{C}$: The degree of shape similarity of the spectra in the positive and negative half-perimeter (Cc close to 1 is similar to the positive and negative half-perimeter profiles of the φ−q ranges, and close to 0 indicates a huge difference):(5)$$Cc=\frac{\sum xy-\sum x\sum y/n}{\sqrt{\left[\sum x^{2}-{\left(\sum x\right)}^{2}/n\right]\left[\sum y^{2}-{\left(\sum y\right)}^{2}/n\right]}},$$
where x, y indicate the average discharge within the positive and negative half-week of the phase window, respectively.

## 3. Experimental Setup

Currently, the existing devices for detecting PD are classified according to the occurrence of different phenomena, often accompanied by ultrasonic waves, pulse signals, gas, light, and thermal energy, resulting in various detection methods such as UHF pulse current and chemical detection. In laboratory and field monitoring, coupling capacitance, UHF, ultrasonic, chemical, and other types of local discharge detection methods are usually used to obtain the PD signal according to the signal characteristics of the law of type identification and localization of PD. Among them, the UHF method is a particularly efficient PD detection method. It has a more significant advantage than other PD detection methods in wisdom network fusion [25,26,27], cost, anti-interference ability, positioning ability, portability, etc. Therefore, this experiment is based on the UHF method.

### 3.1. Experimental Platform Design

The laboratory uses a GIS PD fault simulation system containing five single PD mode models: a built-in non-PD high-voltage transformer, gas insulation can, UHF probe, discharge model, PD measurement coupling capacitor, square wave injection port, and other parts. It can simulate five types of PD: tip discharge, particle discharge, surface discharge, air gap discharge, and suspension discharge. The system can arbitrarily set the discharge type in the charged working mode; control the starting voltage, extinguishing voltage, and discharge intensity of the discharge signal; and test each discharge type by the mature coupling capacitance method and UHF method. The experimental platform is shown in Figure 1 below. During the experiment, the GIS PD fault simulation system sets the discharge model artificially. The voltage is slowly increased until the PD signal appears on the PD instrument. The discharge intensity is gradually increased within a specific range to control the discharge intensity, and the voltage is reduced to zero at the end of the test. The function of the square wave injection port is only used to calibrate the partial discharge detection system using the coupling capacitance method. During the PD test, it is suspended, does not come into contact with the inside, and does not participate in the PD test.

The specific calibration process is to apply appropriate discharge pulses through the standard pulse generator. The calibration is completed when the discharge (pC) value to be calibrated equals the value selected by the calibration pulse generator. Generally, 50 pC and 100 pC standard pulses are injected.

The UHF method of detection with more vital anti-interference ability and higher sensitivity is used. Also, the GDJF-2006S digital PD detection system connected to the coupling capacitor is used to measure the PD signal; the system has more vital anti-interference ability, a detection sensitivity of 0.1 pC, a dynamic range greater than 80 dB, and a measurement bandwidth of 3 dB under the bandwidth range of 10 kHz–1 MHz.

### 3.2. Sensitivity Test Platform Design

According to the Q-GWD UHF method, the PD technical detector was specified for testing sensitivity, and a standard UHF sensor was first selected. The sensor has an average effective height of 16.5 mm and a minimum effective height of 6.7 mm in the 300–1500 MHz frequency band, as shown in Figure 2 below.

The equivalent height, which is the ratio of the amplitude of the sensor’s output voltage to the amplitude of the electric field strength of the incident electromagnetic wave, describes the coupling performance of a sensor [28]. M. D. Judd [29,30] et al. pioneered the use of a gigahertz transverse electromagnetic (GTEM) test cell to measure the equivalent height of a UHF sensor to characterize the sensitivity of the sensor. This has been recommended and used by domestic and foreign power systems and sensor manufacturers. Currently, the performance of UHF sensors in China’s power industry standards is based on GTEM test cell. The measurement of the equivalent height of the sensor needs to borrow the transfer function $H_{ref}$ of the reference sensor. The voltage response value of the sensor under the pulse electromagnetic field within the frequency band range of 300–1500 MHz in the GTEM cell is measured, and the effective height of the tested sensor is calculated according to the following formula (6). Then the average value in the frequency band of 300–1500 MHz is calculated to obtain the average effective height of the tested sensor.
(6)$$H_{sensor}=\frac{U_{ms}}{U_{mr}}\times H_{ref},$$
where $U_{ms}$ and $U_{mr}$ are the measured voltage responses of the measured sensor under testing and the reference sensor, respectively, to the pulsed electric field at the location of the sensor in the GTEM cell.

A calibrated transient electric field is established in the GTEM cell, and the UHF sensor is placed at the opening of the GTEM cell and connected to the designed GIS PD detection system, as shown in Figure 3 below.

## 4. System Design

### 4.1. Hardware Design

A highly representative experiment is established to ensure that the investigation is universal. Traditional PD detection is mainly through the sensor receiving the PD signal through the high-speed acquisition card. Then it processes the data to generate PRPD, PRPS, and other plots in the background and discriminates the type of PD through the plots. However, when the PD signal is collected, there will be a large amount of noise, affecting the PD type’s discrimination. The PD signal will be drowned in severe cases, so noise reduction processing is needed.

For the typical interference of PD, many measures have been taken [31], such as the use of a dedicated independent power supply for power supply interference and cable avoidance of crossover; a little grounding of the experimental circuit for the interference of the grounding system; and grounding of all the surrounding metal objects for the space electromagnetic interference. Despite all the measures taken, the detection is still subject to many internal and external signal disturbances. In this experiment, noise is filtered out in hardware as much as possible, and the signal acquisition and processing device is designed with a mixture of filters to achieve noise reduction, including band-pass filters (300–800 MHz and 1100–1500 MHz), high-pass filters, low-pass filters, and FPGA digital filters that are dynamically adjustable in the range of 1–50 MHz. In addition to this, several other measures have been taken:(1)Dual-channel quadrature down-converter

Conventional signal processing using single down-conversion, the local oscillator signal, and UHF signal mixing will lead to spectral aliasing; the use of dual quadrature down-conversion, the input of the two in-phase and quadrature signals, and the local carrier signal in-phase and quadrature signals multiplied, respectively; the multiplied signals after the addition of the signal in order to obtain the up-converted signals; subtraction operation to obtain the signal of the down-conversion [32]; as shown in Figure 4 below.

(2)Dual-channel data acquisition

The amplified signal is divided into two by the power divider: one way is connected to the frequency mixer and low-pass filter, and the other way is connected to the band-pass filter and detector, which is passed into the AD by analog-to-digital conversion and finally proposed to PRPD and PRPS, as shown in Figure 5 below. Compared with the traditional single-channel acquisition, dual-channel can achieve complementary advantages. Single frequency mixing is used to select the required frequency band through the adjustment of the local oscillator; the advantage is good anti-interference, but it has low detection channel and poor sensitivity in a low-noise environment. The single detector has the advantage of a wide band and high sensitivity when the interference is minor. The disadvantage is that it is easy to ignore the noise in the range of this frequency band. The design uses dual-channel data acquisition through the combination of two ways to achieve complementary advantages of good anti-interference performance and high sensitivity. The designed signal acquisition and processing module is shown in Figure 6 below.

### 4.2. Noise Reduction Algorithm

The PRPD-based noise reduction algorithm is the noise reduction process through PRPD rather than directly through the PD signal. The interfering signal is converted into a pulse by PRPD, achieved by the pulse signal’s effective signal and pulse width. The effective signal is used for periodic filtering, and the pulse width is used for specified non-periodic filtering. Two filtering algorithms are combined to achieve the masking effect: one is used to detect the periodic signal and mask it, and the other to hide the non-periodic signal with the specified pulse width.

For periodic interference, the effective signal of its pulse is also periodic to identify the elimination of interference. The specific steps are counting between each valid signal. As long as it is within a particular range, it is an interference signal directly shielding elimination. The spacing of eight cycles is calculated and averaged to obtain the period filtering parameters, and the marker signal in the cycle is generated. For non-periodic interference, the pulse width is specified according to the sampling rate, and the signal is fed back and masked. The details are shown in Figure 7.

## 5. Experimental Testing

The equipment was connected in the laboratory, as shown in Figure 8 below. The PD simulation system was turned on, but no PD fault model was set; the detected spectrum is shown in Figure 9 below. The simulation system was powered off, and the PRPS spectrum did not change. No PD signals were injected, but there was a pulse in the spectrum, which is obviously noise. The presence of noise interfered with the usual acquisition and analysis of the partially weak PD signal, which was easily drowned in the noise and also affected the system’s sensitivity.

### 5.1. Noise Reduction Treatment

PRPD data were imported, PRPD was converted into pulses, and the pulse signal’s effective signal and width were extracted.

First, the periodic interference was filtered. The experiment set the periodic signal parameter to 500, considered the signal at each effective signal interval count of 500 ± 80 as the interference signal, and masked it, where 500 is the average calculation of eight cycles of spacing and 80 is the value of dynamic adjustment, according to the actual shielding effect of adjustment. Finally, in filtering non-periodic interference, the experiment was set in the pulse width of the parameter 8; 8 ± 2 pulse width signal was specified as the interference signal, shielded, and delayed to obtain the filtered signal, where 8 was calculated according to the sampling rate to accept, 2 was the value of dynamic adjustment to obtain the optimal value. The waveform of the noise signal before and after the PRPD filtering process was obtained through simulation, as shown in Figure 10 below. The signal-to-noise ratio was tested before and after using the PRPD filtering algorithm, as shown in Figure 11 below. Figure 12 shows the PRPS mapping after the secondary noise reduction.

The signal-to-noise ratio is calculated as
(7)$$SNR=20\;\log\frac{V_{s}}{V_{n}},$$
where $V_{s}$ is the amplitude of the signal and $V_{n}$ is the amplitude of the noise.

The *SNR* using only conventional noise reduction is *SNR* = 20 log2.5 = 7.96 dB.

The *SNR* after the PRPD filtering algorithm is *SNR* = 20 log5 = 13.98 dB.

The experiment proves that after using this noise reduction algorithm, the signal-to-noise ratio of the device was significantly improved.

### 5.2. Identification of PD Type

Five PD types were simulated separately by the PD simulation system, and the PRPS profiles of various kinds of devices after the noise reduction treatment were recorded relative to the typical profile characteristics [33,34]. Figure 13 below shows the PRPS profiles obtained by simulating each discharge type through the PD simulation system. Figure 14 shows the PRPS map obtained by the digital PD at this time. Because the digital PD instrument uses only internal synchronization, the phase is different from the plots simulated by the designed PD analog system, but it conforms to the discharge characteristics, which are used here as a comparison.

(1) The characteristics of the tip discharge are more discharges, small dispersion of discharge amplitude, and uniform time interval. The polarity effect of the discharge is obvious and usually appears only in the negative half of the corresponding week of the operating frequency.

The PRPS pattern detected by the simulated tip discharge experiment is shown in Figure 13a below, which is consistent with the characteristics. The PRPS map obtained from the digital PD instrument at this time are shown in Figure 14a below.

(2) The characteristics of the particle discharge are the wide distribution of discharge amplitude, unstable interval, inconspicuous polarity effect, and distribution of discharge signal throughout the entire working frequency cycle.

The PRPS pattern detected by the simulated particle discharge experiment is shown in Figure 13b below, which is consistent with the characteristics. The PRPS map obtained from the digital PD instrument at this time are shown in Figure 14b below.

(3) The air gap discharge is characterized by a low number of discharges, low period repeatability, and scattered amplitude, but the discharge phase is more stable and has no apparent polarity.

The PRPS pattern detected by the simulated air gap discharge experiment is shown in Figure 13c below, which is consistent with the characteristics. The PRPS map obtained from the digital PD instrument at this time are shown in Figure 14c below.

(4) The characteristics of the suspension discharge are the stable discharge pulse amplitude, the same adjacent discharge time interval, and a polarity difference between the positive and negative half-wave detection signals when the suspended metal body is asymmetric.

The PRPS pattern detected by the simulated levitation discharge experiment is shown in Figure 13d below, per the characteristics. The PRPS map obtained from the digital PD instrument at this time are shown in Figure 14d below.

(5) The surface discharge is characterized by large dispersion of discharge amplitude, unstable discharge time interval, and insignificant polarity effect.

The PRPS pattern detected by the simulated surface discharge experiment is shown in Figure 13e below, which is consistent with the characteristics. The PRPS map obtained from the digital PD instrument at this time are shown in Figure 14e below.

The five discharge type mappings are all consistent with the typical PD type characteristics, proving that the noise reduction algorithm does not affect the PD type’s discrimination and retains the PD signal’s characteristics.

### 5.3. Sensitivity Test

Because the output UHF signal waveform is only a response to the current change rate at the discharge source, it is not directly related to the change in the charge amount. The apparent discharge in traditional measurement is due to the charge change on the surface of the equipment caused by partial discharge, so there is no direct relationship between the UHF signal and apparent discharge [35], and the sensitivity cannot be expressed by pC. The system’s sensitivity was further tested according to relevant research and power industry standards [33,36]. The designed signal acquisition and processing module was connected to the UHF sensor placed in the GTEM cell opening, and the output voltage amplitude of the calibration source was adjusted so that the detection system reliably reflects the calibration signal with a signal-to-noise ratio of not less than 2. The minimum transient electric field strength peak $E_{im_{i}n}$ that can be detected by the observation device is the point where the minimum transient electric field strength peak is the system’s sensitivity [37]. The sensitivities before and after the secondary noise reduction using the PRPD noise reduction algorithm were tested separately and are shown in Table 1 below.

The sensitivity of the PD detection system is 1.8 V/m when tested in the GTEM cell with conventional noise reduction processing only and 1.138 V/m when tested with the PRPD noise reduction algorithm with secondary noise reduction processing. The smaller the minimum transient electric field strength peak, the better the sensitivity [38], and it is obvious from Table 1 that the PD detection system meets the requirements of the Chinese Q-GWD UHF method PD charged technical detector specification that the signal-to-noise ratio of the monitoring device (including the sensor) is not less than two times when detecting the transient electric field strength peak of 7 V/m in the GTEM cell. The system sensitivity is significantly improved after the PRPD secondary noise reduction process.

After using this noise reduction algorithm, the device was compared with a digital PD detection system (coupled capacitor method) while connected to a PD simulation system. The thresholds at which the PD signals can be monitored by the two monitoring devices of each type were tested intermittently twenty times, and the average value was taken, as shown in Table 2 below.

The smaller the analog voltage value, the higher the sensitivity. Table 2 shows that the device’s sensitivity is also slightly better than that of the GDJF-2006S digital PD detection system after using the PRPD noise reduction process.

### 5.4. On-Site Testing

As shown in Figure 15 below, tests were performed in the field against a digital localizer with an oscilloscope. The equipment operated normally, with no partial discharges occurring. The site was noisy, with many sources of interference. Noise was evident in the plot before using the secondary filtering algorithm. After using the PRPD secondary filtering algorithm, there was no obvious noise in the map, effectively filtering out the noise in the field, consistent with Figure 12 above.

## 6. Conclusions

This paper has presented a PRPD-based secondary filtering algorithm for the detection of partial discharge in GIS. The algorithm differs from existing noise reduction methods by utilizing PRPD to process the PD signal instead of direct methods. By calculating jitter beyond the threshold, PRPD-related parameters are obtained. Interference signals are identified based on periodic or specified pulse width parameters, and two thresholds are set for filtering. The resulting data are then fed back to the controller to regenerate the mapping. Additionally, a dual quadrature down-converter and dual-channel data acquisition device for PD monitoring were designed. In Section 5, the algorithm is applied to minimize the noise in the laboratory measurements obtained by the ADC while preserving the acquisition of PD signals. Sensitivity tests and simulations of five types of PD were conducted. Experimental results demonstrate that the algorithm does not interfere with the reception of normal PD signals, allowing for accurate insulation status discrimination and improved device sensitivity. The algorithm’s feasibility was verified in the laboratory and the field.

The key to PD noise reduction processing lies in removing noise from measurement data while preserving the characteristic structure of PD signals without distortion. The proposed noise reduction algorithm has proven its feasibility in laboratory settings and effectively reduces noise. It offers significant advantages as a complement to conventional noise reduction processes since it does not require probing the noise source and is straightforward to operate. Future work will involve conducting an adaptive study of PRPD threshold parameters.

## Figures and Tables

**Figure 1 sensors-23-06763-f001:**
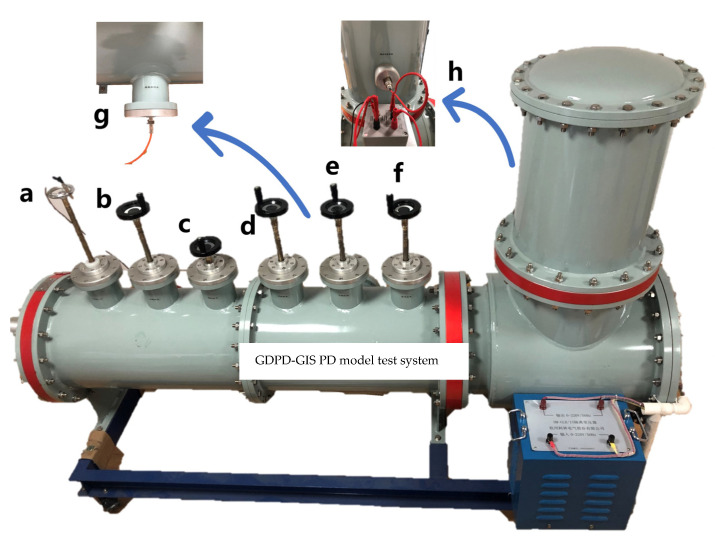
GDPD-GIS PD model test system: (a) square wave injection port; (b) tip discharge; (c) particle discharge; (d) surface discharge; (e) air gap discharge. (f) suspension discharge; (g) UHF sensor; (h) coupling capacitance terminal.

**Figure 2 sensors-23-06763-f002:**
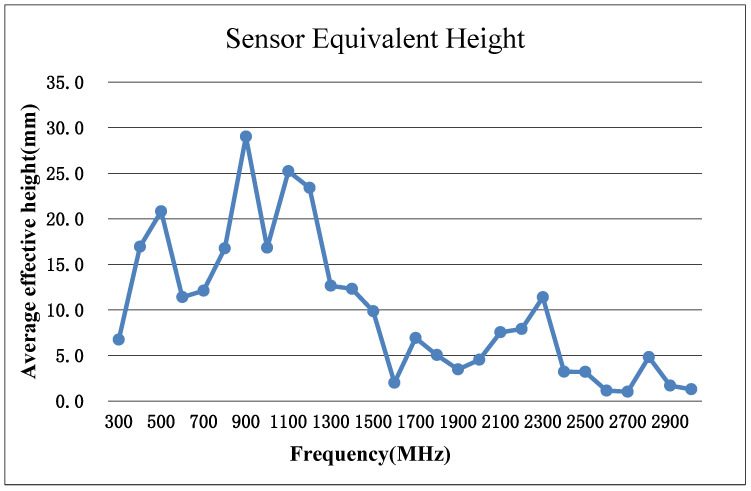
Sensor equivalent height.

**Figure 3 sensors-23-06763-f003:**
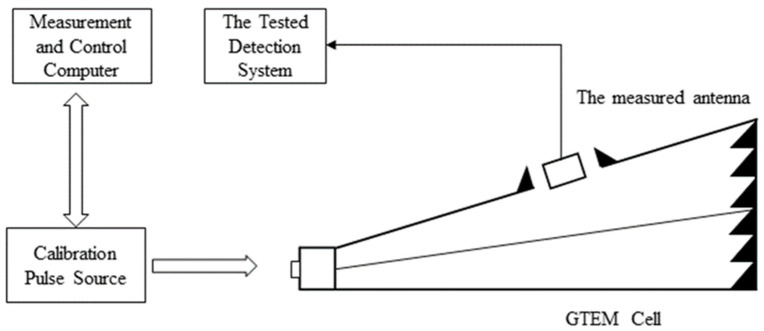
Schematic diagram of the sensitivity calibration platform.

**Figure 4 sensors-23-06763-f004:**
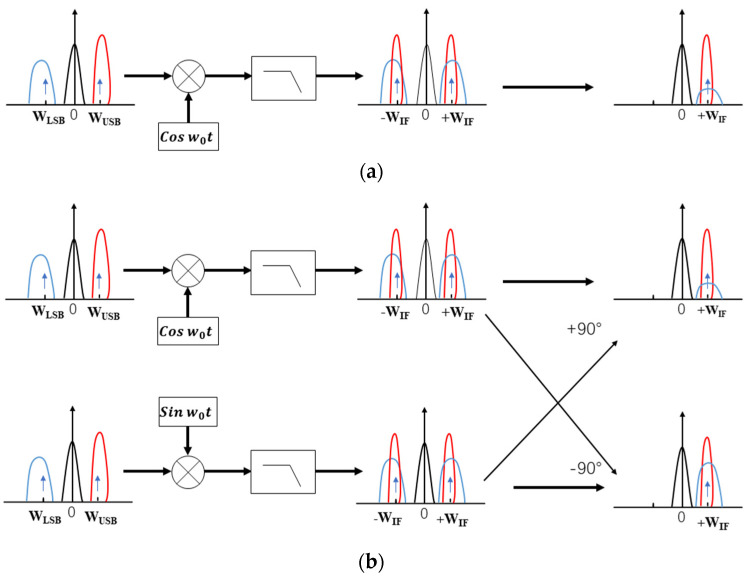
(**a**) Spectrum shift diagram after mixing a local oscillator signal with a UHF signal; (**b**) spectrum shift diagram after two-channel orthogonal.

**Figure 5 sensors-23-06763-f005:**
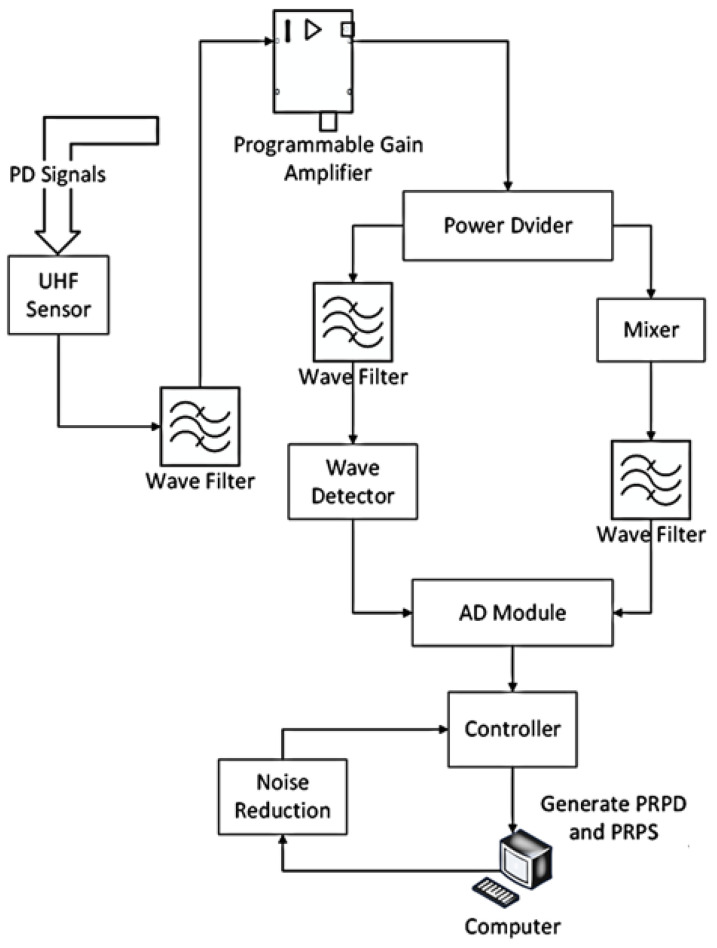
Signal detection flow chart.

**Figure 6 sensors-23-06763-f006:**
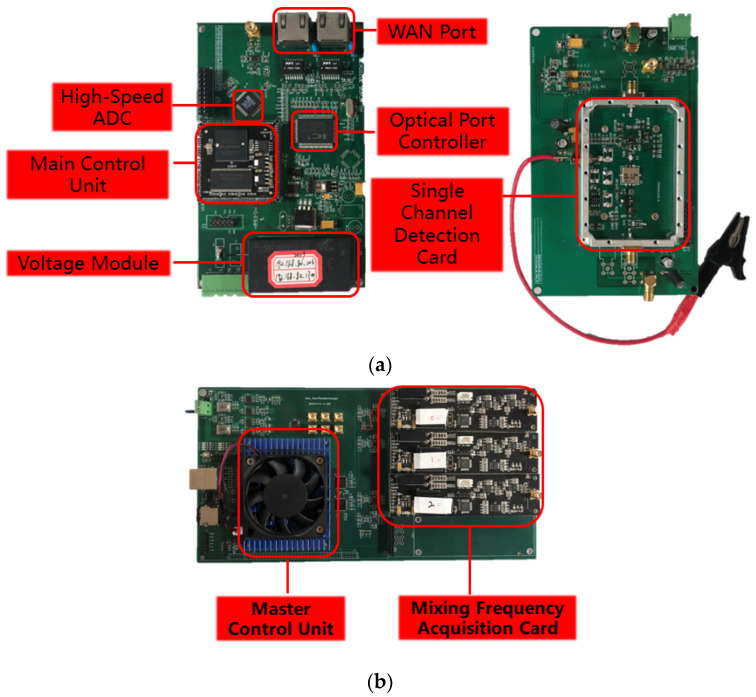
Schematic diagram of signal acquisition and processing part. (**a**) Wave detection module. (**b**) Frequency mixing module. (**c**) Main schematic diagram.

**Figure 7 sensors-23-06763-f007:**
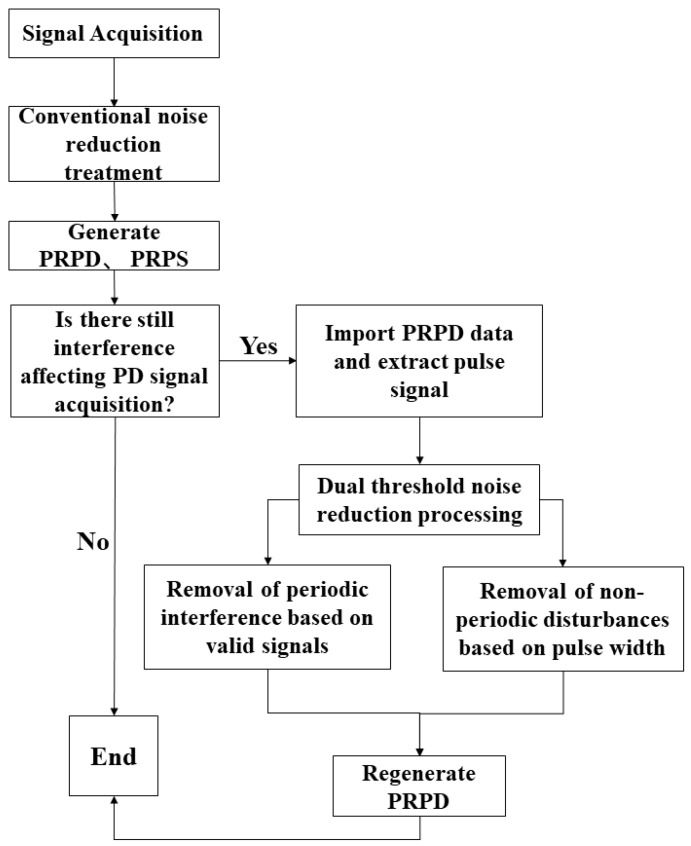
Flow chart of noise reduction algorithm.

**Figure 8 sensors-23-06763-f008:**
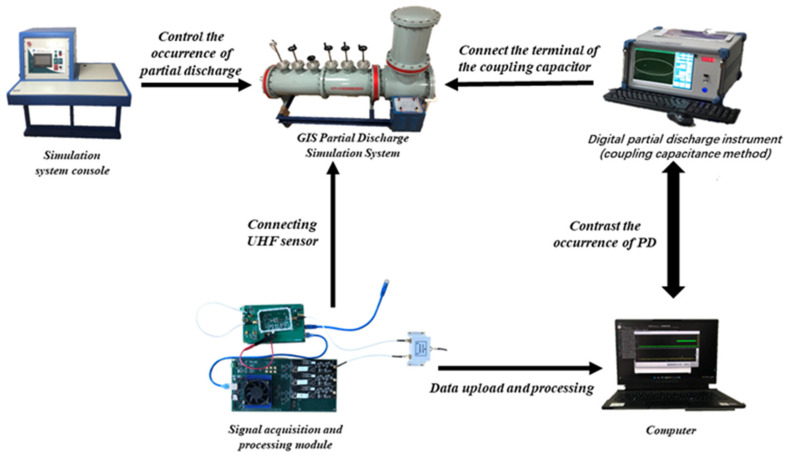
Experimental setup: simulation system console to control the occurrence of GIS PD; simulation system PD; designed signal acquisition and processing module connected to the simulation system’s built-in UHF sensor; data uploaded to the background, against the digital PD instrument connected to the terminal of the coupling capacitor.

**Figure 9 sensors-23-06763-f009:**
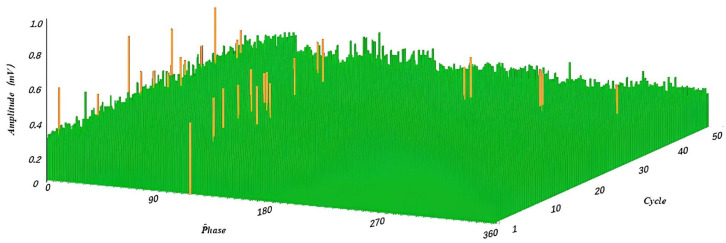
PRPS (orange bumps are noise).

**Figure 10 sensors-23-06763-f010:**
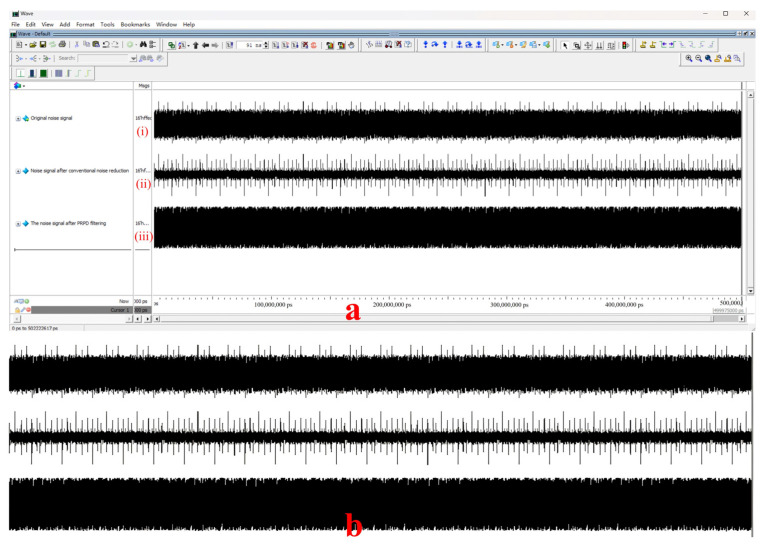
(**a**) Simulated signal plot. (**b**) Signal amplification plot. (i) Original noise signal. (ii) Noise signal after conventional noise reduction, filter, and wavelet algorithm. (iii) The noise signal after PRPD filtering.

**Figure 11 sensors-23-06763-f011:**
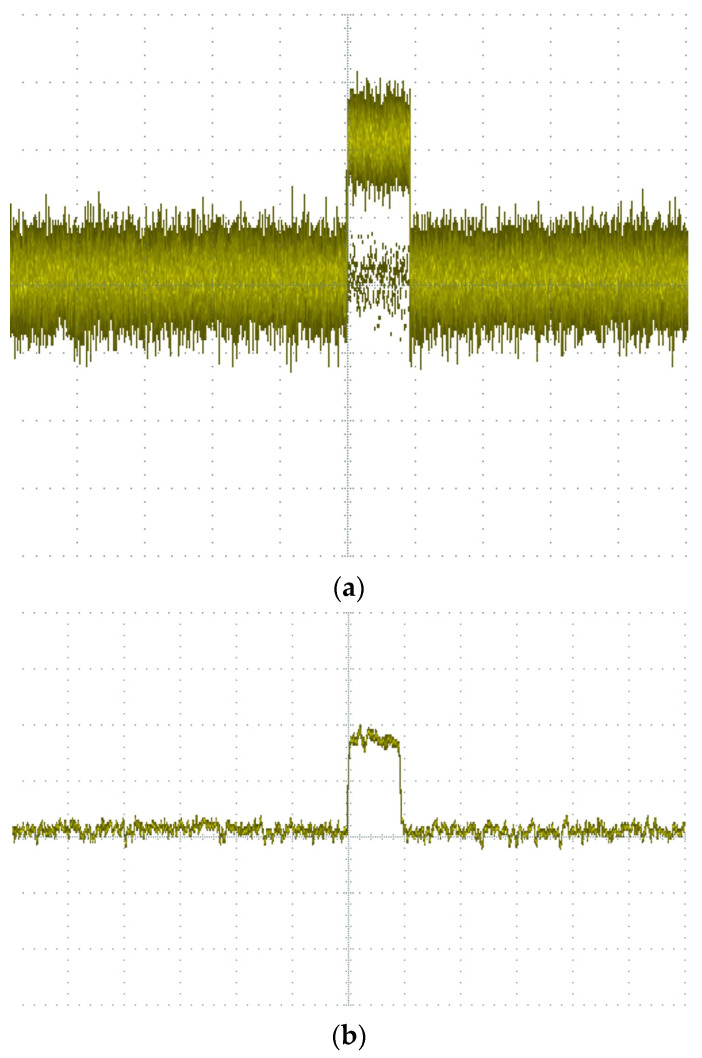
Oscilloscope-acquired waveforms: (**a**) using only conventional noise reduction; (**b**) after PRPD filtering algorithm.

**Figure 12 sensors-23-06763-f012:**
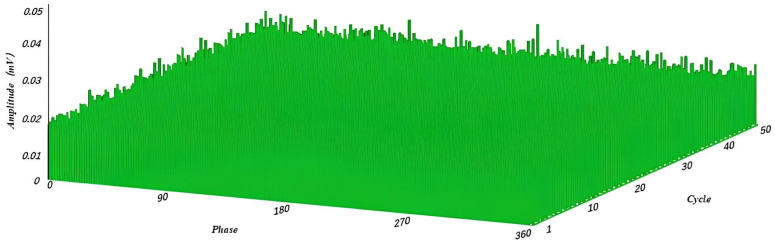
PRPS after noise reduction by PRPD algorithm; at this time there is no significant noise that will affect the PD.

**Figure 13 sensors-23-06763-f013:**
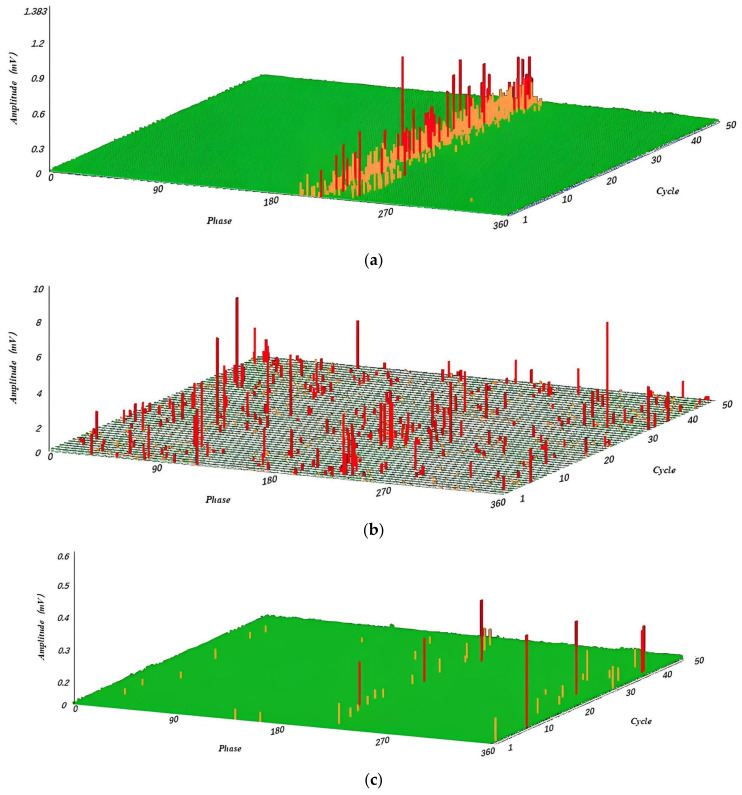
Simulation of the PRPS obtained experimentally: (**a**) Tip discharge appears in simulated system energized to 7.2 kV; (**b**) Particle discharge appears in simulated system energized to 15.2 kV; (**c**) Air gap discharge appears in simulated system energized to 31 kV; (**d**) Suspension discharge appears in simulated system energized to 32.1 kV; (**e**) Surface discharge appears in simulated system energized to 28.5 kV.

**Figure 14 sensors-23-06763-f014:**
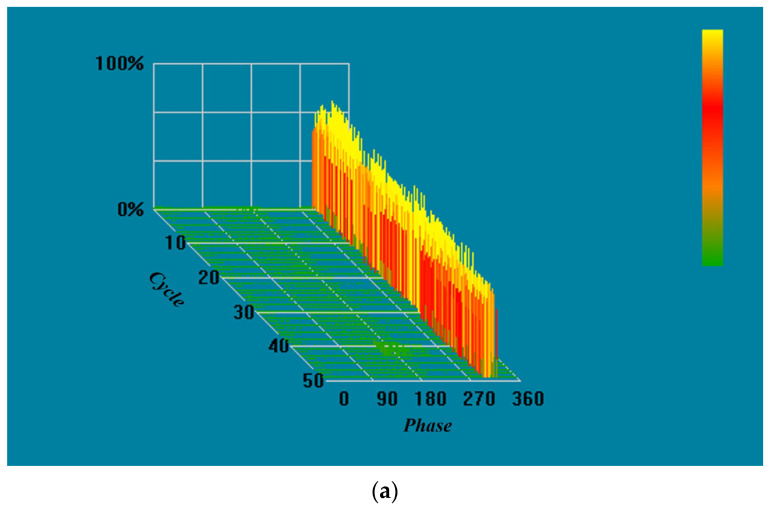
PRPS maps obtained by digital PD instrument: (**a**) tip discharge; (**b**) particle discharge; (**c**) air gap discharge; (**d**) suspension discharge; (**e**) surface discharge.

**Figure 15 sensors-23-06763-f015:**
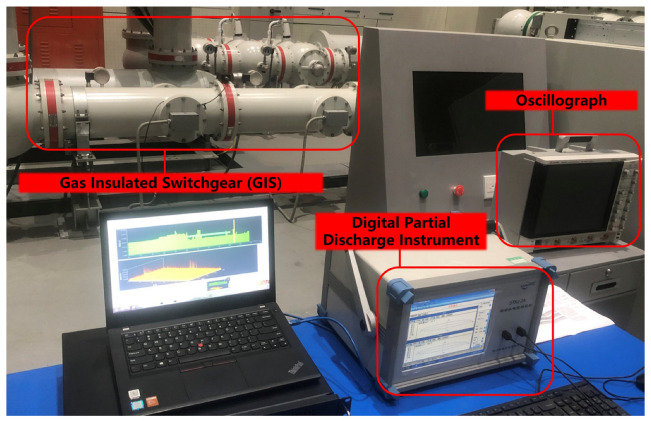
Field test chart.

**Table 1 sensors-23-06763-t001:** System sensitivity.

	Injected Voltage Value Vi (V)	Minimum Transient Electric Field Strength Peak $E_{im_{i}n}$ (V/m)	PD System Response Value dBm/mV
Conventional noise reduction treatment	0	0	0.1042
	0.2	0.409	0.1103
	0.4	0.811	0.1224
	0.6	1.143	0.1306
	0.8	1.8	0.2401
	1.0	1.991	0.3106
System sensitivity	1.8		
Secondary noise reduction using PRPD noise reduction algorithm	0	0	0.1242
	0.2	0.379	0.1307
	0.4	0.759	0.1324
	0.6	1.138	0.2302
	0.8	1.517	0.3455
	1.0	1.897	0.4406
System sensitivity	1.138		

**Table 2 sensors-23-06763-t002:** Analog voltage values (kV) when PD signal appears.

Discharge Type	After Using PRPD Noise Reduction Algorithm (UHF Method)	Digital PD Monitoring System (Coupled Capacitor Method)
Tip Discharge	7.036	7.3
Air Gap Discharge	30.862	30.5
Suspension discharge	31.998	32.5
Particle discharge	15.146	15.9
Surface discharge	28.034	29.5

## Data Availability

Not applicable.

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
