# Peer review of "A PRPD-Based UHF Filtering and Noise Reduction Algorithm for GIS Partial Discharge"

_sensors, 2023, doi:10.3390/s23156763_

Round 1
Reviewer 1 Report
The manuscript needs critical improvements in the literature review and method description.
The literature review is very poor. there are a lot of investigations related to the topic which are not considered. The most of introduction section should be about the weakness of previous research compared with the proposed approach.
The title of the manuscript should consider the main features of your study such as GIS case study and UHF signal investigation.
There are some undefined abbreviations such as ANGPD
The proposed method is not clear compared with previous studies. The advantages are not shown and the comparison results are not provided. Is your method only based on some amateur indices? How much and how it can decrease noise? What is the advantage compared to previous works in different aspects of PD denoising?
The proposed method is not clear and the advantages compared with previous works are not described. It seems that the work is prepared and suitable for a conference paper, not a journal paper.
Reviewer 2 Report
The advantages of the proposed noise reject method are not clear. A comparative table is recommended.
1. In Fig2, what does the Sensor Equivalent Height mean and how does the height was measured?
2. Please add the full name of the GTEM.
3. The explanation of the fig4 is not clear, and the quality of the figures needs to be improved like Fig. 2,4,6,7
4. The experimental platform design mentioned the square wave injection port. What are the square wave parameters for the tests in this article what's the dv/dt frequency.....?
NA
Reviewer 3 Report
The authors present a very interesting research work, in the state of the art and with domain in acquisition electronics, but some points can be improved:
1. Authors must review the entire text.
2. The authors must carry out a methodological improvement in the representation of the evaluated defects. Drawings or photographs must be placed.
3. Authors should reference the sentence ”Among them, the UHF method is a particularly efficient PD detection method. It has a more significant advantage than other PD detection in comparing wisdom network fusion, cost, anti-interference ability, positioning ability, portability, etc.” to justify the statement.
4. What is the sensor gain?
5. Authors must carry out a deep revision and must also improve the quality of all figures, especially figures 6, 9, 10, 11 and 12.
6. In lines 336 and 337 of the manuscript it says that the sensitivity is 1.8 and 1.138 respectively, what are the units of these magnitudes?
7. In Table 2, since all experiments were carried out in conjunction with the coupling capacitor method and its algorithm proved to be capable of detecting discharges at lower inception voltages, for comparison purposes it would be interesting to put the levels in pC of the discharges acquired in Figure 12 and the level in pC of the noise.
8. What is the sensitivity, in pC, of your UHF sensing?
Authors must review the entire text.
Round 2
Reviewer 1 Report
All concerns are answered.
Author Response
Dear Editors and Reviewers:
Thank you for your letter and for the reviewers’ comments concerning our manuscript entitled “A PRPD-based partial discharge filtering and noise reduction algorithm” (ID: sensors-2472768). Based on the reviewers' comments, the title is now changed to "A PRPD-based UHF filtering and noise reduction algorithm for GIS partial discharge".
Thank you very much for your comments and suggestion.
Best regards
Weixing Fang,Guojin Chen,Wenxin Li,Ming Xu,Wei Xie,Chang Chen,Wanqiang Wang,Yucheng Zhu
07 25,2023
Reviewer 2 Report
Most of the questions have been addressed
Minor editing of English language required
Author Response
Dear Editors and Reviewers:
Thank you for your letter and for the reviewers’ comments concerning our manuscript entitled “A PRPD-based partial discharge filtering and noise reduction algorithm” (ID: sensors-2472768). Based on the reviewers' comments, the title is now changed to "A PRPD-based UHF filtering and noise reduction algorithm for GIS partial discharge".
Thank you very much for your comments and suggestions, slight adjustments have been made to the English language.
Best regards
Weixing Fang,Guojin Chen,Wenxin Li,Ming Xu,Wei Xie,Chang Chen,Wanqiang Wang,Yucheng Zhu
07 25,2023
Reviewer 3 Report
This reviewer congratulates the authors for the manuscript and the review performed.
Author Response

(The authors gave the same response as above.)
